# The Trade-Offs and Synergies of Ecosystem Services in *Pinus massoniana* Lamb. Plantations in Guangxi, China

Rongjian Mo [1,2,†], Yongqi Wang [1,2], Yanhua Mo [1,2,†], Lu Li [1,2] and Jiangming Ma [1,2,3,*]

1    Key Laboratory of Ecology of Rare and Endangered Species and Environmental Protection,
     Guangxi Normal University, Ministry of Education, Guilin 541006, China;
     moyanhua2019@mailbox.gxnu.edu.cn (Y.M.)
2    Guangxi Key Laboratory of Landscape Resources Conservation and Sustainable Utilization in Lijiang River
     Basin, Guangxi Normal University, Guilin 541006, China
3    Guangxi Key Laboratory of Superior Timber Trees Resource Cultivation, Guangxi Forestry Research Institute,
     Nanning 530002, China
*    Correspondence: mjming03@gxnu.edu.cn
†    These authors contributed equally to this work.

**Abstract:** A scientific understanding of the synergistic and trade-off relationships among ecosystem services (ESs) is essential for maintaining the structure, function, and health of plantation forest ecosystems. This understanding facilitates effective ecosystem management practices, and helps identify the types, intensities, and spatial and temporal distribution characteristics of interactions among ESs, which is critical for regional development planning, ecological supplementation, and the maximization of economic benefits. In this study, we used correlation analysis, bivariate spatial autocorrelation, and hot spot analysis to comprehensively analyze the synergistic and trade-off relationships between ESs in *Pinus massoniana* (PM) plantations in Guangxi Paiyang Forest Farm from 2009 to 2018, across both time and space. The study showed that the ESs in PM plantations in Guangxi Paiyang Forest Farm maintained significant positive correlation (synergy), with a mutually reinforcing relationship among services. Notably, the regulating services shifted from weak synergy to weak trade-off relationships over time. From the bivariate spatial autocorrelation analysis, it is clear that the overall trade-off synergistic relationship among the four ESs is basically consistent with the correlation analysis results. From the distribution of multiple ES hot spots, we determined that the number of small groups that can provide three to four service hot spots in Guangxi Paiyangshan Forest Farm is greater. The spatial distribution of cold–hot spots of various ESs varied, and the distribution of cold–hot spots of supply services and regulation services of carbon sequestration and oxygen release was similar.

**Keywords:** *Pinus massoniana* plantations; forest ecosystem services; trade-off synergy; service cold–hot spot analysis

## 1. Introduction

Forest ecosystems are essential key systems for human beings, providing various services such as timber volume, edible forest by-products, starch and other products, water conservation, biodiversity conservation, and regulation of the atmospheric environment for human life and production [1,2]. Forests serve as the most complete, abundant, and stable carbon reservoir, natural oxygen reservoir, species gene pool, renewable resource reservoir, freshwater resource, and renewable energy reservoir in nature. Forests play a fundamental and critical role in human survival and the realization of sustainable ecological civilization [3]. Additionally, forest ecosystems offer services to humans in terms of medical health care, physical and mental relaxation, tourism, and rest, as well as other cultural experiences. The ecological and economic benefits of forest ecosystems should be comprehensively coordinated, and sustainable management of forest ecosystems should be maintained to

achieve a mutually beneficial situation for both humans and nature. Schwaiger, F. et al. [4] used a forest management model to analyze the trade-offs between ESs and biodiversity indicators in two case study areas in Germany, which influenced decision-making in three scenarios of forest management. Langner, A. et al. [5] used simulated scenario analysis and multi-criteria decision analysis methods to quantify trade-offs among four ESs (timber production, carbon storage, biodiversity conservation, and landslide hazard control) in montane forests to make decisions about expected utility in multi-objective forest management. Dai, E. et al. [6] valued four services of timber volume, carbon storage, water production, and soil conservation in the Ganjiang River basin, quantified their trade-offs and synergies at the county scale and sub-basin scale, and detected synergistic relationships between timber volume and carbon storage, soil conservation, and water production. Wu, W. et al. [7] studied the trade-offs and synergistic relationships between two regulating services, soil organic carbon regulation, total nitrogen regulation, and species diversity, as well as their influencing factors in a mixed *Pinus* sylvestris and *Quercus* sylvestris needle-broad forest, a deciduous broad-leaved southern date palm forest, and a broad-leaved evergreen forest of *Cycad* and *Quercus* sylvestris, using secondary forests at different successional stages in the central subtropical region.

Currently, both domestic and international research on forest multifunctionality has primarily focused on the fundamental meaning of forest multifunctionality [8–10], as well as the evaluation of multiple ESs and their spatiotemporal evolution characteristics [11–13]. However, further research is needed to develop and enhance our understanding of forest ES trade-offs and synergies. In this study, we explored the spatiotemporal relationship characteristics among multiple ESs in PM plantation forests by combining statistical analysis methods, spatial mapping, and bivariate spatial autocorrelation methods. This study is a continuation of the previous assessment of ESs in a *Pinus massoniana* plantation. We utilized the value of ESs in PM plantations in Guangxi Paiyangshan Forest Farm as an indicator, and employed correlation analysis to identify the absolute values of correlation coefficients and the positive and negative directions to determine the types and strength characteristics of relationships among ESs. Additionally, we employed the bivariate spatial autocorrelation method to describe the synergistic and trade-off relationships among ESs in space, and applied the hot spot analysis method to identify cold–hot spot areas of ESs. The spatial quantification and mapping of ESs in PM plantation forests can be useful for revealing scale effects and interactions of ESs, and providing a scientific basis for management decisions in PM plantation forests.

## 2. Materials and Methods

### 2.1. Study Area

The Guangxi Paiyangshan Forest Farm (Figure 1), located in Ningming County, southwest Guangxi, China, near the border with China and Vietnam is the only large state-owned forestry field directly under the Forestry Department of Guangxi Autonomous Region bordering Vietnam, with a total land area of 28,096.29 hm$^2$ and a forest land area of 27,462.28 hm$^2$. The geographical coordinates are 106°30′–107°15′ E, 21°15′–22°30′ N. The southern red loam hilly area in China is a significant timber production region, with approximately 308,000 km$^2$ of planted forests. PM, one of the primary fast-growing production species, is also a critical afforestation and deforestation species in southern China. It is the most widely distributed and largest area timber species in pine [14,15]. PM has outstanding features, including drought tolerance, ridge thinning resistance, strong growing ability, rapid growth, high quality, and versatility [16]. These qualities have great economic and ecological benefits [17,18], making PM an essential component in forestry production and forest restoration and reconstruction in China [19]. In this study, the Paiyangshan Forest Farm PM plantation is in the northern tropical case study area [20], which has an average annual temperature of 21.8 °C, an average annual rainfall of 1250–1700 mm, a rainy season from May to August, and a relative humidity of 82.5%. The annual number of sunshine hours is 1650.3, and the annual evaporation is 1423.3 mm. After years of afforestation,

the native vegetation has been replaced by natural broad-leaved secondary forests and planted forests. The plantation forest tree species are mainly PM, *Eucalyptus robusta*, and *Illicium verum*.

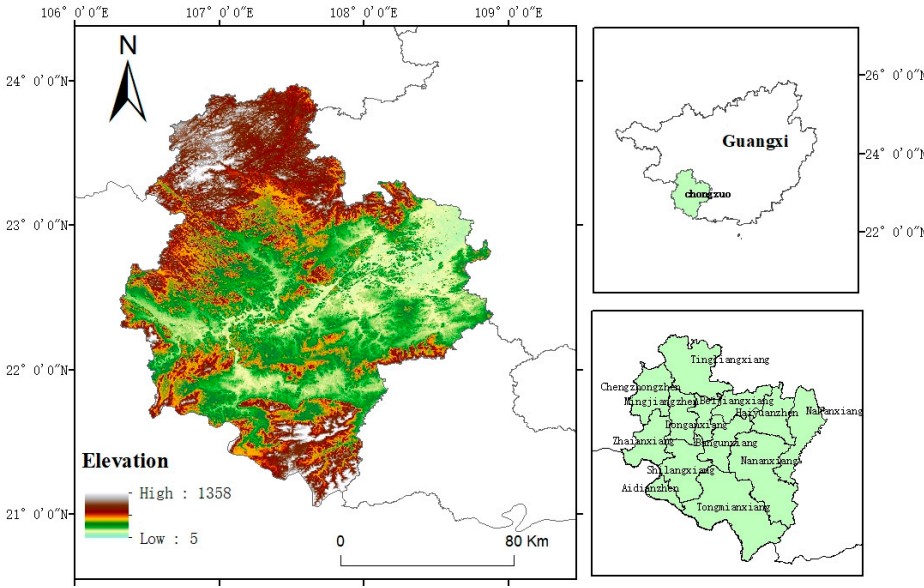

**Figure 1.** Distribution map of study area.

*2.2. Research Methods*

In this study, we employed correlation analysis, bivariate spatial autocorrelation analysis, and hot spot analysis to explore the relationship between the ES values of the PM plantations in Guangxi Paiyangshan Forest Farm [20]. The research framework is illustrated in Figure 2.

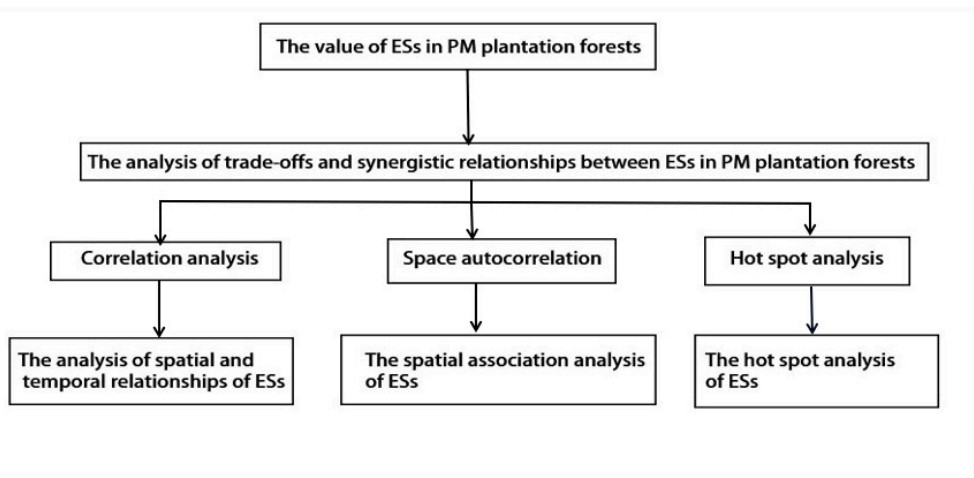

**Figure 2.** Research framework.

2.2.1. Correlation Analysis

The commonly used statistical analysis methods in ES trade-off studies include correlation analysis, root mean square deviation method, regression analysis, and cluster analysis [21,22].

Correlation analysis is the simplest and most effective method to identify trade-offs between ESs. The magnitude of the coefficients that are significantly correlated between ES pairs and the direction of positive and negative correlation are used to determine the

degree and direction of correlation among service pairs. Han, Y. [23] employed simple correlation analysis to calculate the correlation coefficients between production services, water conservation services, carbon sequestration and oxygen release services, environmental purification services, and cultural and recreational services in Xi'an metropolitan area in 1999, 2006, and 2009. The aim was to identify synergies and trade-offs among agricultural ESs. Partial correlation analysis can examine the interrelationships between ES pairs, excluding other factors based on the complex relationships among ESs. Sun, Y. et al. [24] used partial correlation analysis to study the synergistic and trade-off characteristics of water content, soil conservation, net primary productivity of vegetation, and food supply in the Loess Plateau, using Yan'an City as an example. Alamgir, M., et al. [25] used regression analysis and clustering methods to analyze the differences and linkages of multiple ESs among different forest stands.

In this study, a negative correlation coefficient between two ESs that passed the significance test was considered to indicate a trade-off relationship, while a positive correlation coefficient that passed the significance test was considered to indicate a synergistic relationship. Overall, the single services in the study area exhibit spatial heterogeneity, which may lead to both trade-off and synergistic relationships. Therefore, we calculated and tested the correlation coefficients between pairs of four ESs corresponding to small classes in the study area for significance.

In this study, we employed Pearson correlation analysis to investigate the interactions among ESs in PM plantation forests. Pearson correlation analysis is a commonly used method for analyzing relationships among ESs [26,27]. We then used the chart.Correlation function in the R language software to explore the interrelationships among the four major forest services and each sub-forest of the PM plantation, and created a scatter plot matrix. When the correlation coefficient of a pair of ESs is significantly positive, it indicates a synergistic relationship between the services. Conversely, a significantly negative correlation coefficient indicates a trade-off relationship between the services. If the correlation is not significant, then there is spatial compatibility between the services. The strength of the correlation is determined by the absolute value of the correlation coefficient. A correlation coefficient in the range of (0, 0.3) indicates a weak correlation, that in the range of [0.3, 0.5) indicates a moderately strong correlation, and that in the range of [0.5, 1] indicates a strong correlation [28,29].

### 2.2.2. Bivariate Spatial Autocorrelation

To further examine the spatial distribution of synergistic and trade-off relationships among ecosystem services, we assigned a value to each ES in the vector map of small group units. Next, we imported these data into the GeoDa software to analyze the bivariate spatial autocorrelation between pairs of services in the PM plantation forest using the bivariate spatial autocorrelation module. High–high and low–low agglomerations both indicate synergistic relationships, while high–low and low–high agglomerations indicate trade-off relationships [30].

### 2.2.3. Hot Spot Analysis

Spatially cold–hot spots refer to areas with either high or low values for ESs provided by PM plantation forests [31]. In this study, we used the hot spot analysis tool in the ArcGIS spatial analysis module and the Getis-Ord Gi* local statistic to identify cold–hot spots for four services [32]. To determine whether a small class was a hot or cold spot, we used the aggregation index degree parameter $Z(Gi^*)$ and followed the classification criteria described in the study by Wang, B. et al. [33]: if $Z(Gi^*) > 2.58$, the region was classified as a hot spot; if $1.65 < Z(Gi^*) \leq 2.58$, it was a sub-hot spot; if $Z(Gi^*) < -2.58$, it was a cold spot; if $-2.58 \leq Z(Gi^*) < -1.65$, it was a sub-cold spot; and if $-1.65 \leq Z(Gi^*) \leq 1.65$, it was not considered significant.

## 3. Results

### 3.1. Analysis of ES Interactions in PM Plantation Forests

Based on Figure 3, the correlation results of the ecosystem services (ESs) of PM plantations in Guangxi Paiyangshan Forest Farm from 2009 to 2018 can be observed. In 2013, the value per unit area of wood supply, carbon sequestration and oxygen release, and water conservation became more concentrated, while the distribution of soil conservation was more dispersed. In 2018, the data points for water conservation were more concentrated than in 2013, while the data points for other services were more dispersed. From a correlation standpoint, the six groups of services provided by PM plantations in Guangxi Paiyangshan Forest Farm from 2009 to 2018 were all significantly positively correlated, except for carbon sequestration and oxygen release, which were negatively correlated with water conservation in 2018.

Wood supply was strongly positively correlated with carbon sequestration and oxygen release in 2009, and weakly positively correlated with water conservation (correlation coefficient 0.19) and soil conservation (correlation coefficient 0.20). This is similar to the strong synergistic relationship between food provision and carbon sequestration and oxygen release in the value of cropland ESs reported by Tian, Y. [26]. Carbon sequestration and oxygen release were weakly correlated with water conservation (correlation coefficient 0.18) and soil conservation (correlation coefficient 0.17). The correlation between water conservation and soil conservation (correlation coefficient 0.17) was also weak. In 2013, the correlation increased (0.29) and was close to moderate for water conservation and soil conservation, while all other service pairs decreased while maintaining their original correlation. In 2018, the correlation between wood supply and carbon sequestration and oxygen release increased (correlation coefficient 0.68), the correlation between water conservation and soil conservation increased (correlation coefficient 0.48), and the correlation between the other service pairs decreased. There was a negative correlation between carbon sequestration and oxygen release and water conservation, indicating a trade-off between the two. The synergistic relationship between wood supply and carbon sequestration and oxygen release decreased and then increased. The synergistic relationship between water conservation and soil conservation continued to increase, while the synergistic relationship between carbon sequestration and oxygen release and water conservation changed from a synergistic relationship to a trade-off relationship. The synergistic relationship between the other service pairs continued to decrease over the years.

In terms of ES types, from 2009 to 2018, the supply service (wood supply) and regulating service (carbon sequestration and oxygen release) of Guangxi Paiyangshan Forest Farm were in a strong synergistic relationship that first weakened and then strengthened. The synergistic relationship between regulating service (water conservation) and supporting service (soil conservation) changed from weak to moderate synergistic. The synergistic relationship between regulating services (carbon sequestration and oxygen release as well as water conservation) changed from synergistic to trade-off. The relationship between the other service pairs showed a weak synergistic relationship that gradually weakened.

Overall, the service pairs of PM plantation forest at the macroscopic scale of Guangxi Paiyangshan Forest Farm maintained a positive relationship from 2009 to 2018. The supply service and the regulating service showed a strong synergistic relationship, the supply service and the supporting service showed a weak synergistic relationship, and the regulating service and the supporting service showed a weak synergistic relationship. However, the regulating service changed from a synergistic to a weak trade-off relationship among them, and the relationship between water conservation and the supporting service in the regulating service changed from a weak synergistic to a moderate synergistic relationship.

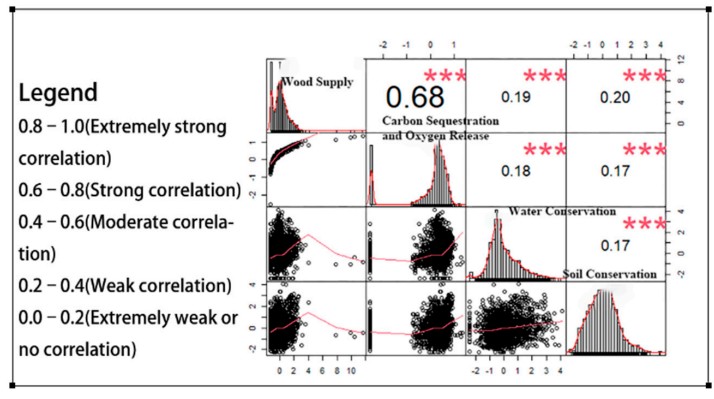

(**a**)

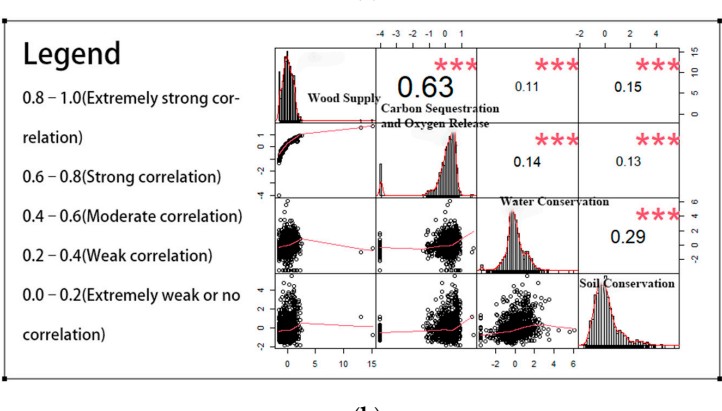

(**b**)

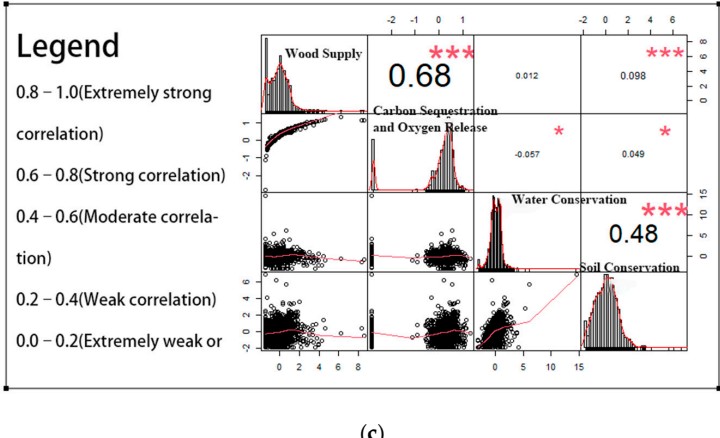

(**c**)

**Figure 3.** A scatter plot matrix was used to show the results of the ES correlation analysis of PM plantation forests in Guangxi Paiyangshan Forest Farm from 2009 to 2018. Pearson correlation coefficients range from −1 to 1, indicating negative and positive correlations, respectively. The strength of correlation increases as the absolute value of the Pearson correlation coefficient approaches 1. Statistical significance is indicated by asterisks, with * representing significance at the 0.05 level, and *** at the 0.001 level. The scatterplot matrices for 2009, 2013, and 2018 are presented as (**a**), (**b**), and (**c**), respectively.

### 3.2. Spatial Association of ESs in PM Plantation Forests

After analyzing the temporal and spatial correlation of ESs, we gained initial insights into the synergistic and trade-off relationship between the ESs of PM plantations in Guangxi Paiyangshan Forest Farm. We determined that the synergy was greater than the trade-off at the scale of the whole forest. Bivariate spatial autocorrelation analysis was conducted on the ESs of PM plantations in Guangxi Paiyangshan Forest Farm, and the spatial distribution

of "HH" type areas was determined to have high spatial correlation of high value of PM plantation function, indicating enhanced synergy with the ESs of PM plantations in adjacent small groups. The "LL" type area is the area with a high spatial correlation of low value of PM plantation function, indicating weakened synergy with the adjacent small group PM plantation ES pairs. The "LH" and "high–low" type areas are small classes with high and low spatial correlation of ES pairs, respectively, indicating a trade-off between ES pairs in adjacent small classes, whereby one service is enhanced or weakened in a small class unit while another function is weakened in an adjacent small class. In other words, one service is enhanced or weakened while another function is weakened or enhanced in the adjacent small group. Due to space limitations, we have only analyzed the spatial correlations between wood supply services and other services here.

Based on Table 1 and Figure 3, the analysis results are significant at a level higher than 95%. As seen in Table 1, the trade-off synergistic relationship of each service pair is generally consistent with the results of the correlation analysis (Figure 3). However, the local autocorrelation index, Moran I, between carbon sequestration and oxygen release and water conservation in 2018 is negative, indicating a weak trade-off, whereas other service pairs are primarily synergistic with each other. Specifically, the synergistic relationship between wood supply and carbon sequestration and oxygen release, water conservation, and soil conservation is spatially robust, with the strongest synergistic relationship being between wood supply and carbon sequestration and oxygen release. The synergistic relationship gradually decreases from 2009 to 2018 with other services. The synergistic relationship between carbon sequestration and oxygen release and soil conservation is initially strong but then decreases. The synergistic relationship between carbon sequestration and oxygen release and water conservation shows a weak synergistic relationship from 2009 to 2013 and a weak trade-off relationship in 2018. The overall spatial relationship between water conservation and soil conservation is weakly synergistic, increasing first and then decreasing. The overall synergistic relationship between multiple service pairs shows a decreasing trend from 2009 to 2018.

**Table 1.** Bivariate local autocorrelation Moran I index of ESs in PM plantations in Paiyangshan Forest Farm.

| ESs Service Pairs | 2009 | 2013 | 2018 |
|---|---|---|---|
| Wood Supply–Carbon Sequestration and Oxygen Release | 0.4248 | 0.2992 | 0.3369 |
| Wood Supply–Water Conservation | 0.1844 | 0.1442 | 0.0393 |
| Wood Supply–Soil Conservation | 0.1720 | 0.1204 | 0.0705 |
| Carbon Sequestration and Oxygen Release–Water Conservation | 0.1710 | 0.1409 | −0.0295 |
| Carbon Sequestration and Oxygen Release–Soil Conservation | 0.1094 | 0.1160 | 0.0266 |
| Water Conservation–Soil Conservation | 0.1716 | 0.2298 | 0.1291 |

The analysis of spatial autocorrelation, with wood supply as the central indicator, and other ESs of neighboring small groups as the corresponding indicators, revealed significant spatial heterogeneity in the trade-off synergistic relationship between wood supply and other services in the PM plantation.

From 2009 to 2018, the synergistic and trade-off areas between wood supply and carbon sequestration and oxygen release were more synergistic than trade-off areas (Figure 4). In 2009, the synergistic enhancement areas were mainly located in the Beishan branch, Gongwu branch, a small portion of small classes in Honghu branch, and the northwestern part of Pucheng branch. However, these areas decreased significantly by 2013. Specifically, the synergistic enhancement areas in Beishan branch shifted to the north, and the synergistic enhancement areas in Gongwu branch and Pucheng branch decreased. By 2018, the synergistic enhancement area of Gongwu branch decreased significantly, while the synergistic enhancement area of Nianke branch and Honghu branch increased slightly. Additionally, the synergistic weakening area was mainly concentrated in the west and east of Gongwu branch, the middle of Nianke branch, Honghu branch, Nalai branch,

and Dawangshan branch in 2009. The synergistic weakening area in the western part of Dawangshan branch, Nalai branch, and Gongwu branch decreased, but Beishan branch increased. The number of synergistic weakening areas increased in Gongwu branch and Pucheng branch and decreased in Beishan branch in 2018, with little change in the number of synergistic weakening areas from 2009 to 2018. Moreover, the weighted areas were mainly distributed in Gongwu branch and Honghu branch in 2009. However, the weighted areas were sporadically distributed in major sub-fields in 2013. In 2018, the weighted areas increased, mainly distributed in Honghu branch and Nianke branch. The number of "high–low" and "low–high" balance areas remained close over the years.

From 2009 to 2018, the synergistic and trade-off areas between wood supply and water conservation were more synergistic than trade-off areas (Figure 5). In 2009, the areas of synergistic enhancement were mainly located in the southern Gongwu branch and the western and eastern parts of the Pucheng branch. However, the synergistic enhancement area significantly reduced in 2013, particularly in the Gongwu branch. In 2018, the synergistic enhancement area of the Pucheng branch decreased while the southern Gongwu and Beishan branches saw a significant increase. Conversely, the areas where the synergy weakened in 2009 were mainly concentrated in the Honghu branch, Nianke branch, the northwestern part of the Nalai branch, and the Dawangshan branch. This area significantly reduced in 2013 and only remained in the Beishan branch, southwestern part of Nianke branch, northwestern part of Honghu branch, and Gongwu branch. However, the synergistic weakening area increased again in 2018, particularly in the Honghu branch and Dawangshan branch, while it decreased in the Beishan branch. Over the years, the number of regions showing synergistic changes decreased and then increased. In 2009, the "low–high" trade-off area was mainly located in the southern part of Gongwu branch, Honghu branch, and the southern part of Pucheng branch, while the "high–low" trade-off area was mainly located in the Honghu branch, Beishan branch, the northwest end of Gongwu branch, and Nianke branch. In 2013, the number of trade-off areas decreased, with "high–low" trade-off areas mainly in the northwestern part of the forest and "low–high" trade-off areas in the southeastern part. In 2018, the number of weighing areas increased, with the "high–low" weighing areas mainly existing in the northern part of the entire forest, such as Honghu branch and Dawangshan branch, while the "low–high" weighing areas were located in the southern part of Beishan branch, Gongwu branch, and the southern part of Pucheng branch. The number of trade-off areas has fluctuated over the years, and the number of "low–high" trade-off areas has consistently exceeded the number of "high–low" trade-off areas.

From 2009 to 2018, the synergistic and trade-off areas between wood supply and water conservation were more synergistic than trade-off areas (Figure 6). In 2009, the synergistic enhancement areas were mainly located in the northwestern part of the Gongwu and Pucheng branches, but this area significantly reduced in 2013 and was only distributed in the northwestern part of the Pucheng branch. However, in 2018, the number of synergistic enhancement areas increased, particularly in the northwestern part of the Pucheng, Gongwu, Nianke, and northern part of Daeangshan branches. The areas of weakened synergy were distributed across all major branches in 2009 but decreased in 2013, mainly in the Beishan and Nianke branches, the western part of the Honghu branch, and the northeastern part of the Dawangshan branch. However, in 2018, the area of weakened synergy increased, mainly in the Honghu, Nianke, and Dawangshan branches. In 2009, the "low–high" weighing area was distributed across all branches, while the "high–low" weighing area was mainly located in the Honghu, Beishan, and Gongwu branches. The number of weighing areas decreased in 2013, and the "low–high" weighing areas were mainly located in the northern and southern parts of Gongwu and Pucheng branches, while the "high–low" weighing areas were located in the western and middle parts of the forest. However, in 2018, the number of weighing areas increased, and the "low–high" weighing areas were mainly located in the Gongwu, Dawongshan, and Pucheng branches, while the "high–low" weighing areas were located in the Gongwu, Honghu, and Pucheng

branches. The number of trade-off areas decreased and then increased over the years, and the "high–low" trade-off area was consistently higher than the "low–high" trade-off area.

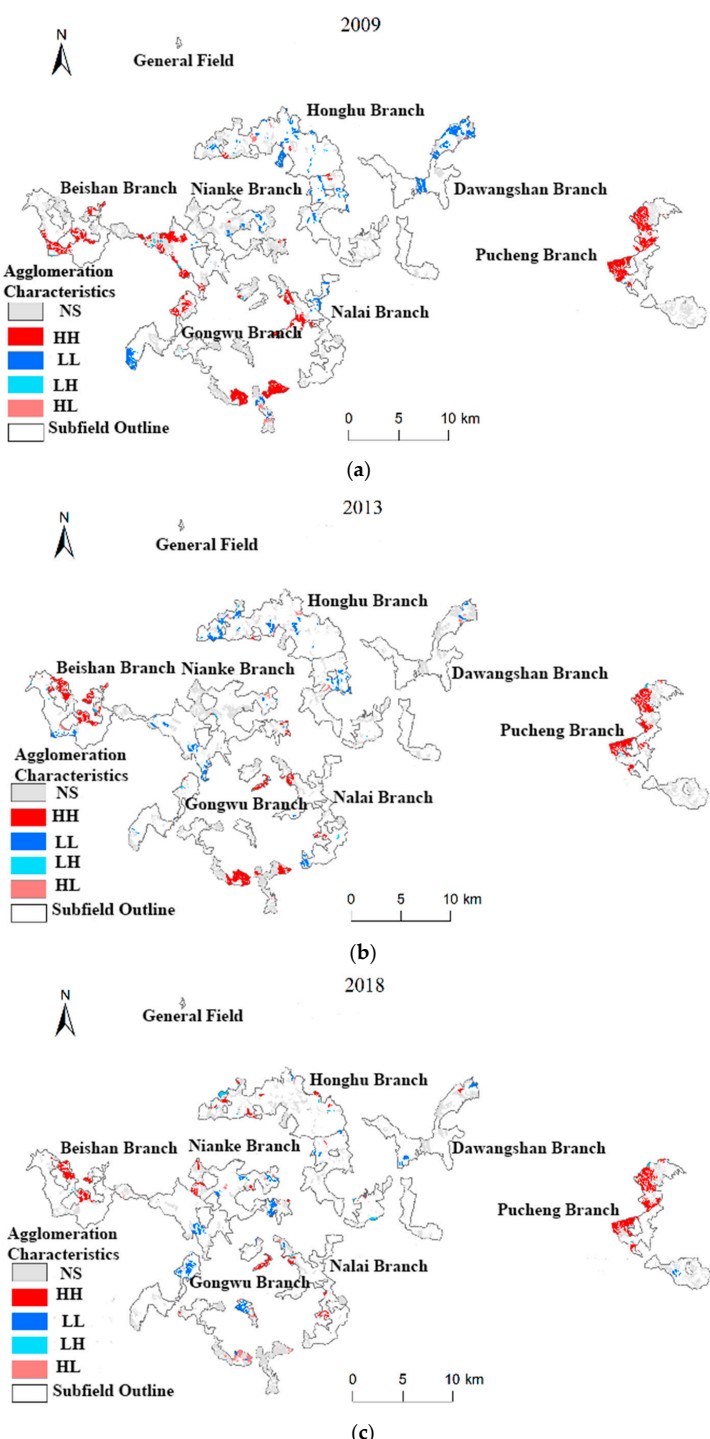

**Figure 4.** This is a Local LISA plot that shows the relationship between wood supply and carbon sequestration and oxygen release services in the PM plantations of Guangxi Paiyangshan Forest Farm from 2009 to 2018. The analysis used the following indicators: NS (not significantly clustered), HH (high–high clustering), LL (low–low agglomeration), LH (low–high agglomeration), HL (high–low agglomeration). The plot includes the following LISA maps: (**a**) wood supply and carbon sequestration and oxygen release services in 2009; (**b**) wood supply and carbon sequestration and oxygen release services in 2013; and (**c**) wood supply and carbon sequestration and oxygen release services in 2018.

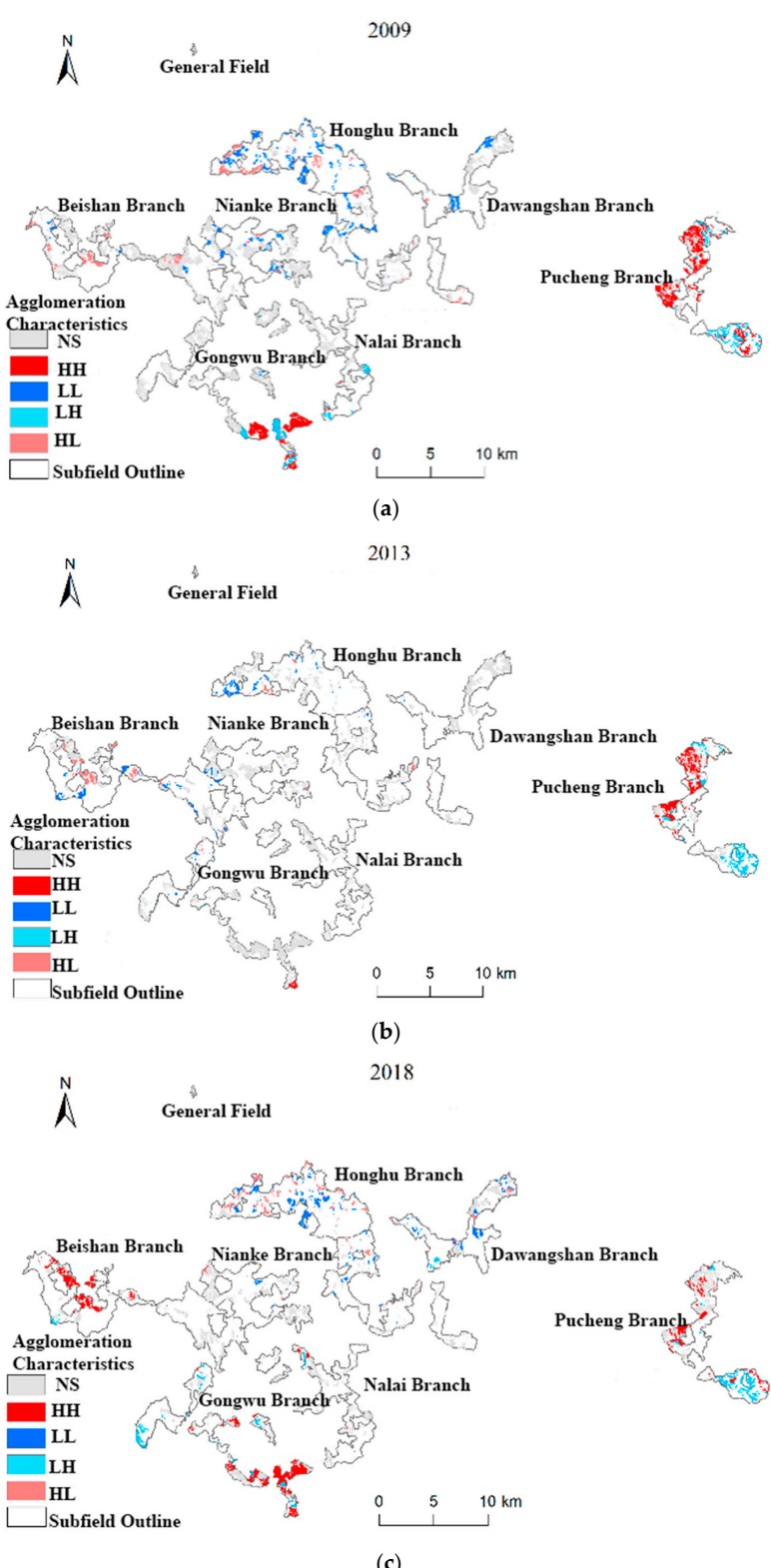

**Figure 5.** The Local LISA map shows the relationship between wood supply and water conservation services in PM plantations in Paiyangshan Forest Farm from 2009 to 2018. The map is categorized into NS (not significantly clustered), HH (high–high clustering), LL (low–low agglomeration), LH (low–high agglomeration), and HL (high–low agglomeration). We present three LISA maps: (**a**) the map of wood supply and water conservation services in 2009, (**b**) the map of wood supply and water conservation services in 2013, and (**c**) the map of wood supply and water conservation services in 2018.

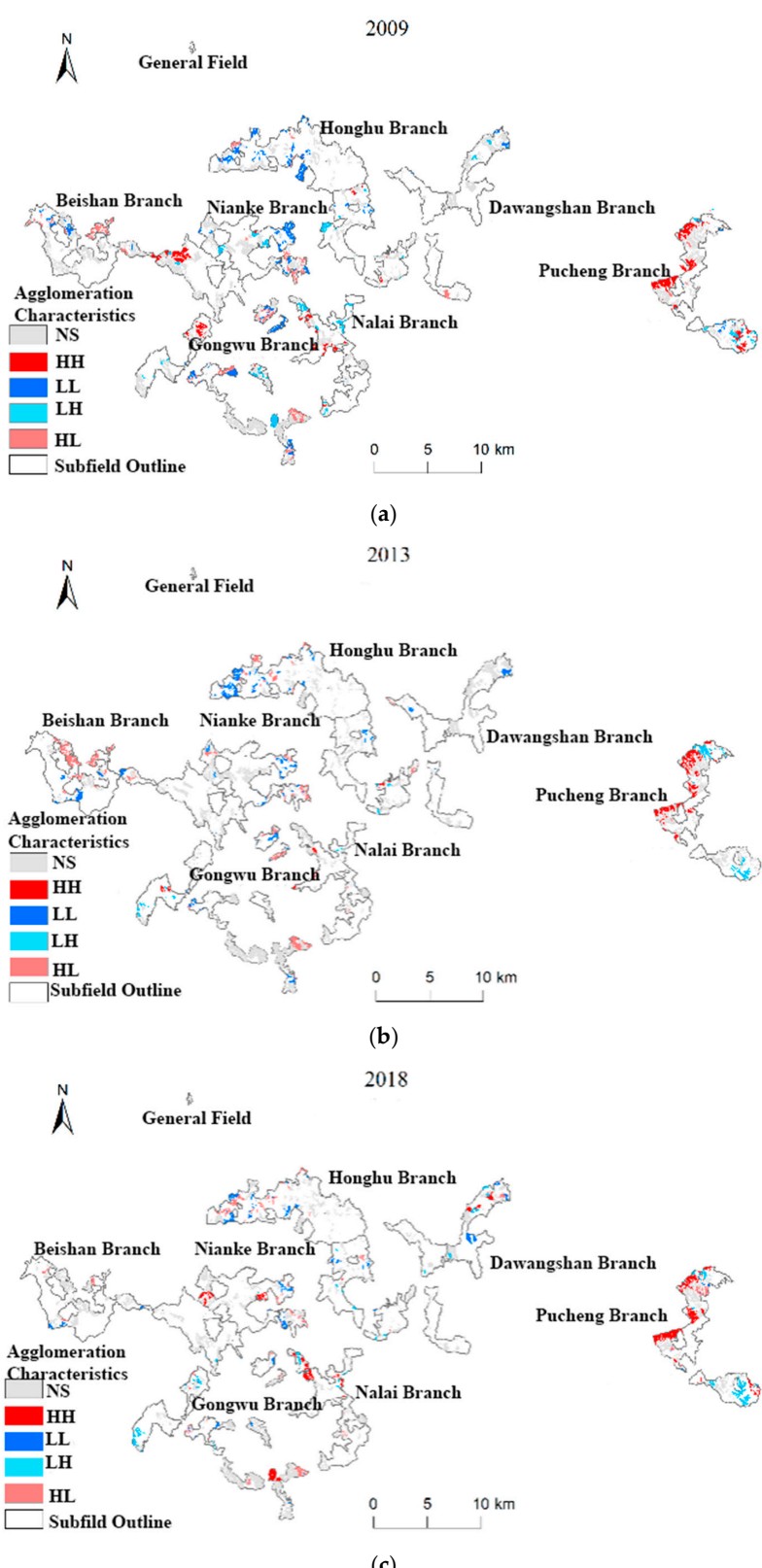

**Figure 6.** These Local LISA plots show the relationship between wood supply and soil conservation services in PM plantations in Paiyangshan Forest Farm from 2009 to 2018. The plots are categorized into NS (not significantly clustered), HH (high–high clustering), LL (low–low agglomeration), LH (low–high agglomeration), and HL (high–low agglomeration). The LISA maps for wood supply and soil conservation services are presented for three years: (**a**) 2009, (**b**) 2013, and (**c**) 2018.

### 3.3. Single ES Space Cold Hot Spot Analysis

The spatial variation and clustering characteristics of ES values were analyzed using hot spot analysis. The number and proportion of small groups of cold and hot spots were counted for each service value in PM plantation forests.

According to Table 2, the percentage of cold and hot spots of the four ESs in Guangxi Paiyangshan Forest Farm changed from 2009 to 2013. The service with the largest percentage of hot spots and sub-hot spots in small groups shifted from carbon sequestration and oxygen release to water conservation. The service with the smallest percentage of cold spots and sub-hot spots from 2009 to 2018 remained carbon sequestration and oxygen release. Among the four services, carbon sequestration and oxygen release were the most prominent in Guangxi Paiyangshan Forest Farm.

**Table 2.** Statistics on the number of small classes of cold hot spots of ESs in PM plantations in Guangxi Paiyangshan Forest Farm in 2009–2018.

| | ES | | Wood Supply | Carbon Sequestration and Oxygen Release | Water Conservation | Soil Conservation |
|---|---|---|---|---|---|---|
| 2009 | Hot Spots and | Number | 621 | 894 | 532 | 674 |
| | Sub-hot Spots | Proportion | 28.33% | 40.78% | 24.27% | 30.75% |
| | Cold Spots and | Number | 722 | 489 | 1042 | 629 |
| | Sub-cold Spots | Proportion | 32.94% | 22.31% | 47.54% | 28.70% |
| 2013 | Hot Spots and | Number | 450 | 499 | 412 | 423 |
| | Sub-hot Spots | Proportion | 31.80% | 35.27% | 29.12% | 29.89% |
| | Cold Spots and | Number | 441 | 349 | 541 | 510 |
| | Sub-cold Spots | Proportion | 31.17% | 24.66% | 38.23% | 36.04% |
| 2018 | Hot Spots and | Number | 486 | 583 | 752 | 517 |
| | Sub-hot Spots | Proportion | 28.45% | 34.13% | 44.03% | 30.27% |
| | Cold Spots and | Number | 396 | 338 | 649 | 486 |
| | Sub-cold Spots | Proportion | 23.19% | 19.79% | 38.00% | 28.45% |

By combining Table 2 and Figure 7, we can observe that in 2009, the number of hot and cold spots of wood supply service value in small groups was 621 and 722, respectively. Hot spots and sub-hot spots were more concentrated in spatial distribution, mainly in the central and eastern part of Beishan branch, Gongwu branch, northern and western part of Pucheng branch. Cold spots and sub-cold spots were mainly distributed in the central and western part of Gongwu branch, Nianke branch, Honghu branch and Dawangshan branch. In 2013, the number of hot and cold spots of small classes of wood supply service value amounted to 450 and 441, respectively, with an increase in the proportion of hot spots and a decrease in the proportion of cold spots. Compared with 2009, the number of hot spots decreased mainly in Gongwu branch, while it increased in Beishan branch. The number of small groups of cold spots mainly increased in Gongwu branch as well as Pucheng branch and decreased in Nianke branch. The number of small classes of hot and cold spots of wood supply service value quantity in 2018 was 486 and 396, respectively. Hot spots and sub-hot spots were mainly distributed in the northern part of Beishan branch, some small classes in the northern part of Honghu branch as well as the northern part of Pucheng branch, and the hot spot distribution area was reduced. Cold spots and sub-cold spots were more dispersed, mainly distributed in the southwest end of Beishan branch, the middle of Gongwu branch, the south end of Honghu branch, the south of Pucheng branch, and the west of Dawangshan branch. The distribution of cold spots significantly reduced compared with 2013. From 2009 to 2018, the percentage of hot spots of wood supply services first increased and then decreased, the percentage of cold spots continued to decrease, the area of lower wood supply services continued to decrease, and the distribution pattern of the whole forest value volume changed to high east–west and low middle.

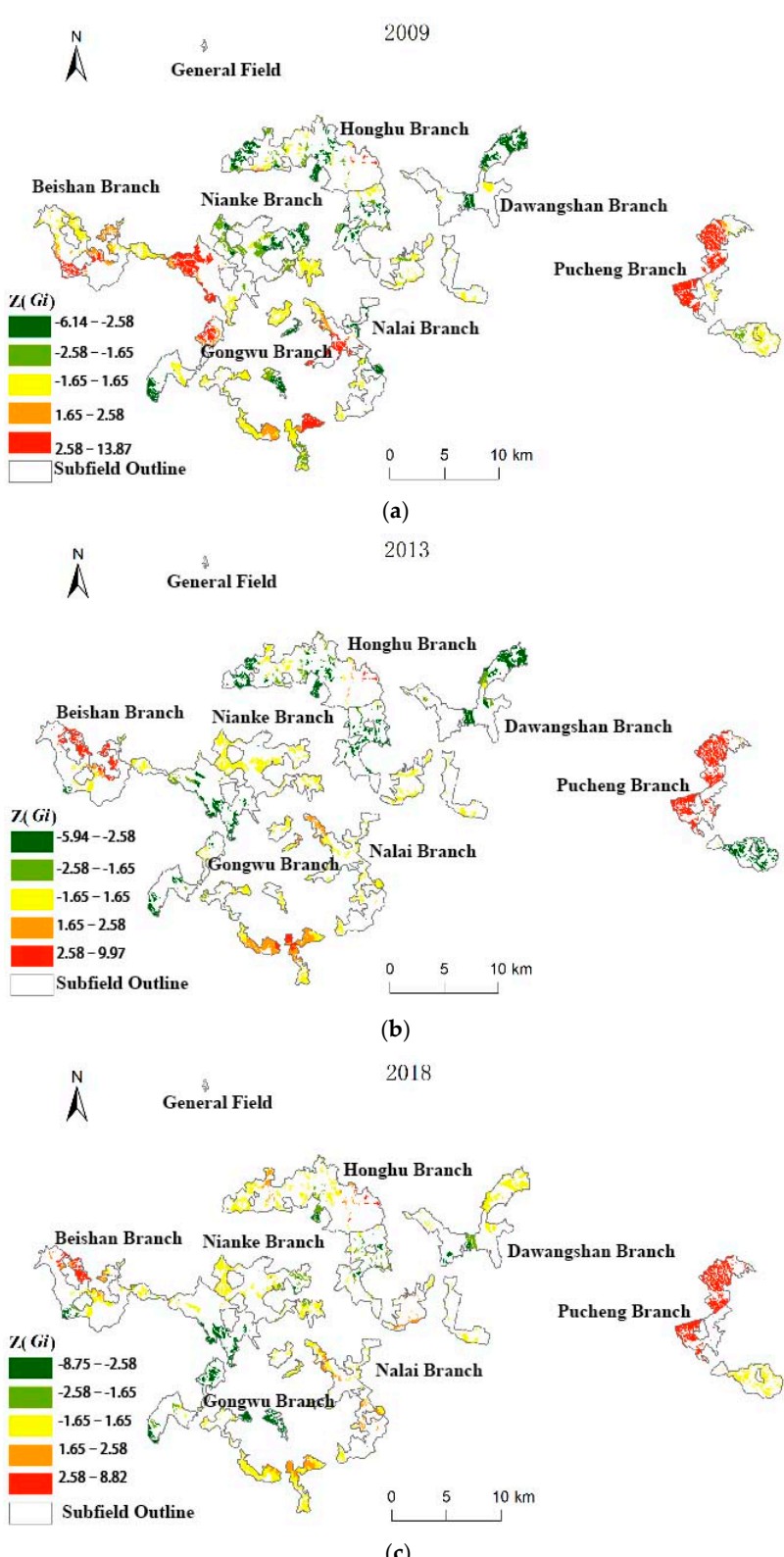

**Figure 7.** This figure shows the distribution of cold–hot spots of wood supply for PM plantations in Paiyangshan Forest Farm from 2009 to 2018, with Z(Gi*) as the aggregation index degree parameter. The color scheme used is red for hot spot regions, orange for sub-hot spot regions, yellow for insignificant regions, dark green for cold spot regions, and light green for sub-cold spot regions. Panel (**a**) displays the distribution of cold–hot spots in 2009, panel (**b**) displays the distribution in 2013, and panel (**c**) displays the distribution in 2018.

By combining Table 2 and Figure 8, we can observe that in 2009, there were 894 hot spots and 489 cold spots in the number of small groups with service values of carbon sequestration and oxygen release. The hot spots and sub-hot spots were concentrated in the central part of Beishan branch, the northwestern and eastern parts of Gongwu branch, and the northern part of Pucheng branch. Meanwhile, the cold spots and sub-cold spots were mainly distributed in the southern and southwestern ends of Gongwu branch, the northeastern part of Dawangshan branch, the northwestern part of Honghu branch, and Nalai branch. In 2013, the number of small groups of hot–cold spots for the value of carbon sequestration and oxygen release services decreased to 499 and 349, respectively, with the percentage of hot spots decreasing and the percentage of cold spots increasing. Compared with 2009, the hot spots decreased in the southern part of Beishan branch but increased in the northern part of Beishan branch and the northwestern part of Gongwu branch. Meanwhile, the number of small groups of cold spots mainly in the western part of Gongwu branch and Dawangshan branch decreased, but that in the southern part of the Gongwu branch increased. In 2018, there were 583 hot–cold spots of the value of carbon sequestration and oxygen release services, with 338 being cold spots. The percentage of hot and sub-hot distribution areas decreased, but the distribution range changed. The hot spot area of the value of carbon sequestration and oxygen release appeared in Honghu branch, and the distribution range of cold spots and sub-cold spots changed significantly. In 2013, the percentage of small classes of cold spots was mainly concentrated in the eastern part of Gongwu branch and Nianke branch, and there were no cold spots in Honghu branch and Beishan branch. However, the percentage of cold spots for carbon sequestration and oxygen release services increased and then decreased from 2009 to 2018, while the percentage of hot spots continued to decrease. Overall, from 2009 to 2018, the distribution pattern of the entire forest value volume changed to high east–west and low middle.

By combining Table 2 and Figure 9, we can observe that in 2009, the number of hot–cold spots of water conservation service value in small groups was 532 and 1042, respectively. The distribution of cold–hot spots was relatively concentrated, with hot spots and sub-hot spots mainly distributed in the east and south of PM plantations in Guangxi Paiyangshan Forest Farm (south of Gongwu branch), and cold spots and sub-cold spots mainly distributed in the northwest of PM plantations in Guangxi Paiyangshan Forest Farm (north of Beishan branch and Nianke branch, northwest of Narai branch and Honghu branch, and west and northeast of Dawangshan branch). In 2013, there were 412 and 541 small classes of hot–cold spots in the value volume of water conservation services. The percentage of hot spots increased while the percentage of cold spots decreased. Compared with 2009, the hot spots were mainly located in the south of Gongwu branch. The number of small groups of cold spots mainly decreased in the northwest of Gongwu branch, Nianke branch, the northwest of Nalai branch, and Dawangshan branch. In 2018, there were 752 and 649 small classes of hot–cold spots in the value volume of water conservation services, and the percentage of hot spots continued to increase while the percentage of cold spots continued to decrease. Hot spot areas appeared in the southwest of Beishan branch and Gongwu branch, and hot spot areas of water conservation value volume appeared. The distribution range of cold spots and sub-cold spots changed considerably, with the Beishan branch transforming into a hot spot area, the northwestern part of the Gongwu branch not being significant, and cold spots appearing in the Dawangshan branch. The percentage of hot spots for water conservation services continued to increase while the percentage of cold spots continued to decrease from 2009 to 2018. The distribution pattern of cold–hot spots of the entire forest value shifted from high in the southeast and low in the northwest to high in the west of the southeast and low in the north of the center.

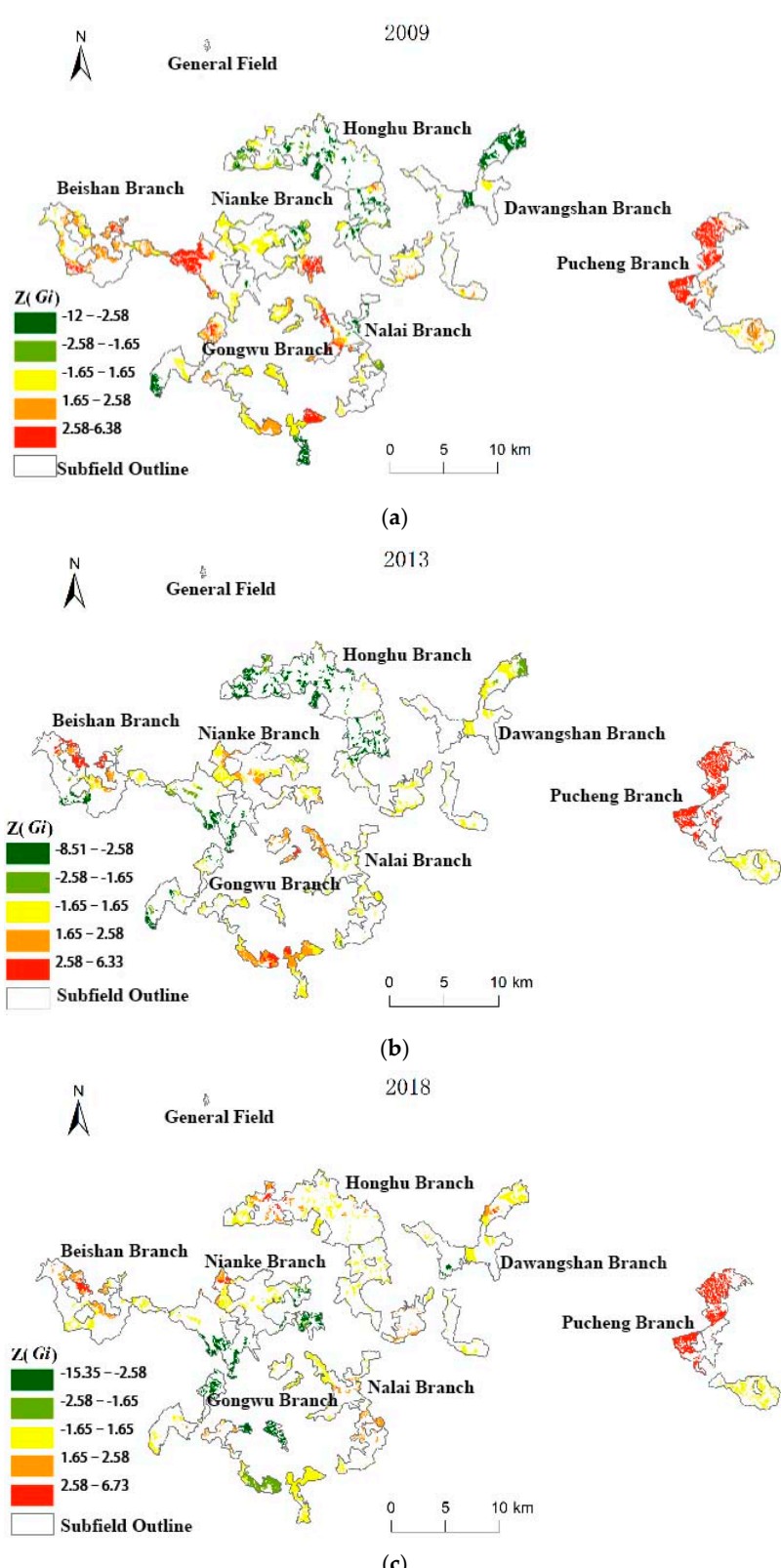

**Figure 8.** Distribution of cold–hot spots of carbon sequestration and oxygen release in PM Plantations in Paiyangshan Forest Farm from 2009 to 2018. Z(Gi*) is the aggregation index degree parameter. Red indicates a hot spot region, orange indicates a sub-hot spot region, yellow indicates an insignificant region, dark green indicates a cold spot region, and light green indicates a sub-cold spot region. The distribution of cold–hot spots of carbon sequestration and oxygen release is shown for the years 2009 (**a**), 2013 (**b**), and 2018 (**c**) in the following figures.

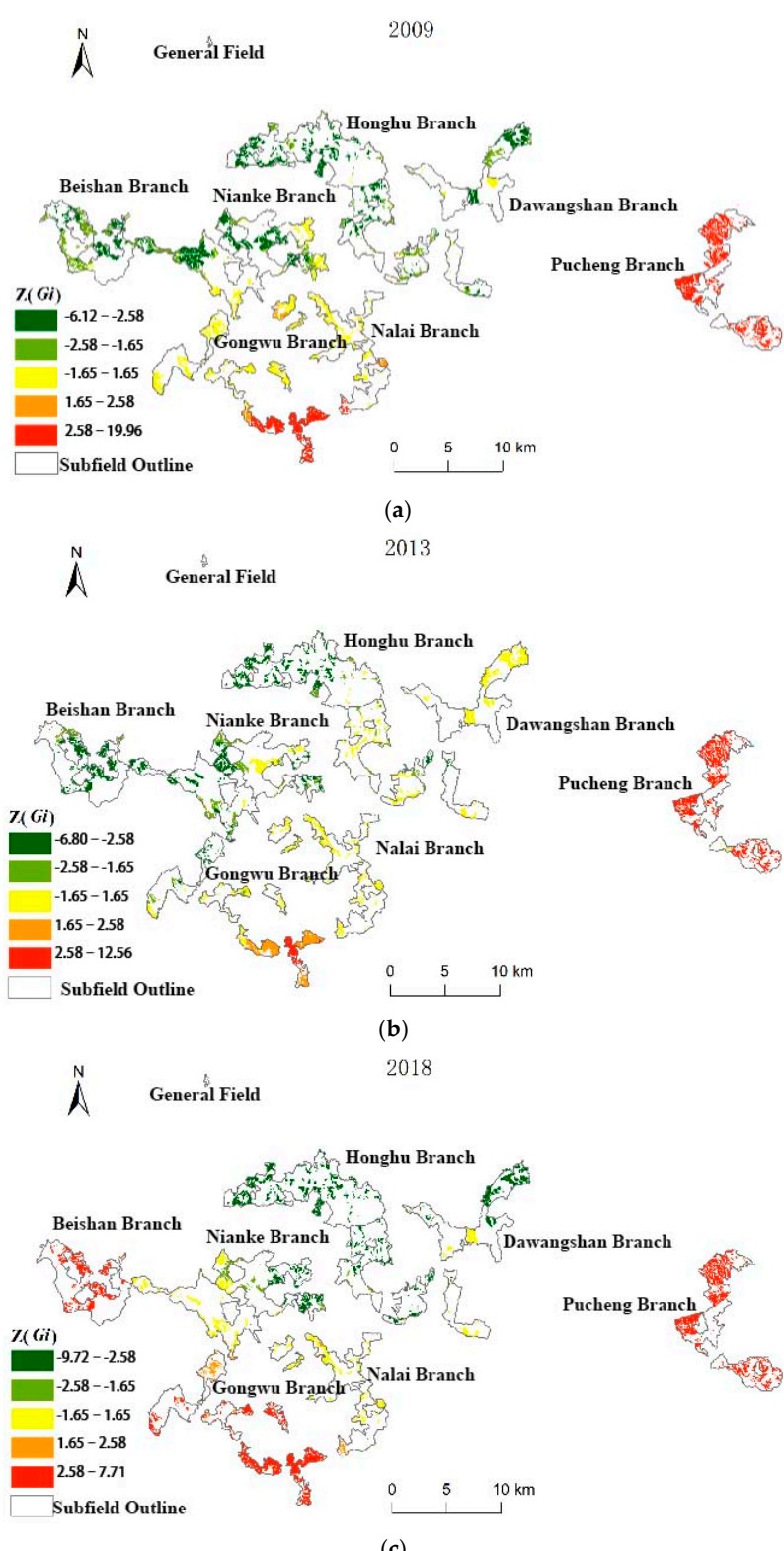

**Figure 9.** Distribution of cold–hot spots of water conservation in PM Plantations at Paiyangshan Forest Farm, 2009–2018. The aggregation index degree parameter Z(Gi*) was used to analyze the spatial pattern of water conservation services. Red indicates hot spot regions, orange indicates sub-hot spot regions, yellow indicates insignificant regions, dark green indicates cold spot regions, and light green indicates sub-cold spot regions. The following maps show the distribution of cold–hot spots of water conservation services in 2009 (**a**), 2013 (**b**), and 2018 (**c**), respectively.

By combining Table 2 and Figure 10, we can observe that in 2009, the number of small groups of hot–cold spots for soil conservation service value was 674 and 629, respectively. Hot spots and sub-hot spots were mainly distributed in the northwestern and central parts of Gongwu branch, the northwestern part of Nalai branch, and Pucheng branch, while cold spots and sub-cold spots were mainly distributed in the northern part of Beishan branch, the eastern and northwestern end of Nianke branch, the central and southern parts of Gongwu branch, and Honghu branch. In 2013, the number of hot and cold spot small classes of soil conservation service was 423 and 510, respectively, with the percentage of hot spots decreasing and the percentage of cold spots increasing. Compared with 2009, hot spots were mainly in the northwestern and central part of Gongwu branch and the northwestern part of Nalai branch, while their numbers increased in the central part of Nalai branch and Pucheng branch. The number of small groups of cold spots increased mainly in the eastern part of Beishan and Dawangshan branch, and decreased in Gongwu branch. In 2018, the number of hot–cold spot small classes for the value of soil conservation services was 517 and 486, respectively, with an increase in the percentage of hot spots and a decrease in the percentage of cold spots. The hot spot distribution area changed less, and decreased in Nalai branch. The cold spot distribution area was widely reduced, distributed in the eastern part of Nianke branch and Honghu branch, and other former cold spot areas became hot and cold insignificant areas. The proportion of hot spots in soil conservation services decreased and then increased, while the proportion of cold spots increased and then decreased from 2008 to 2019. The distribution pattern of cold–hot spots of the entire forest value shifted from an overall staggered pattern of cold–hot spots to high in the east and central part and low in the north.

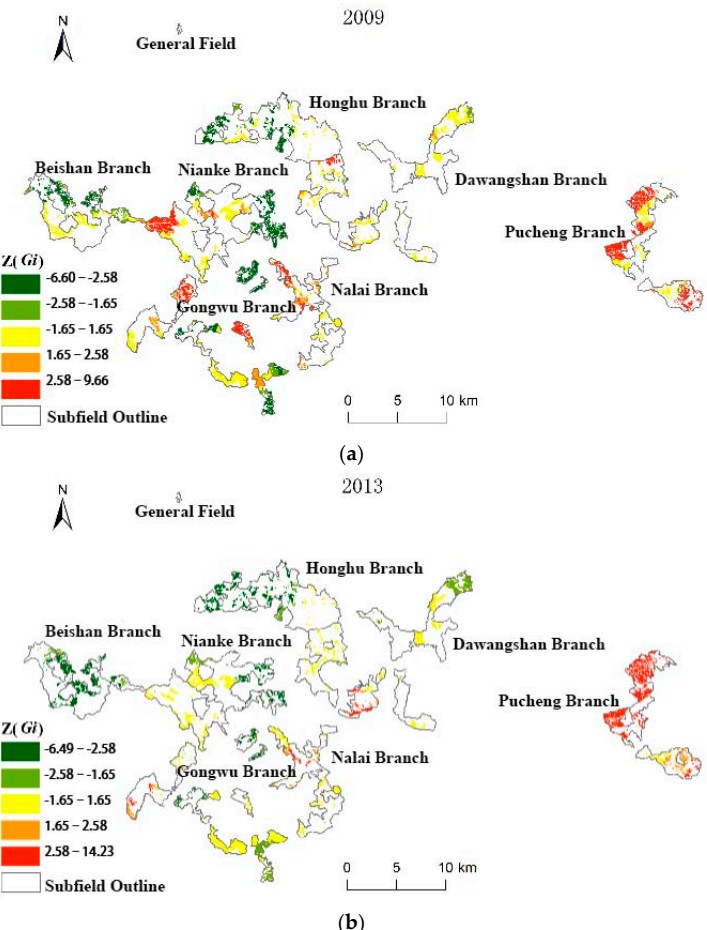

**Figure 10.** *Cont.*

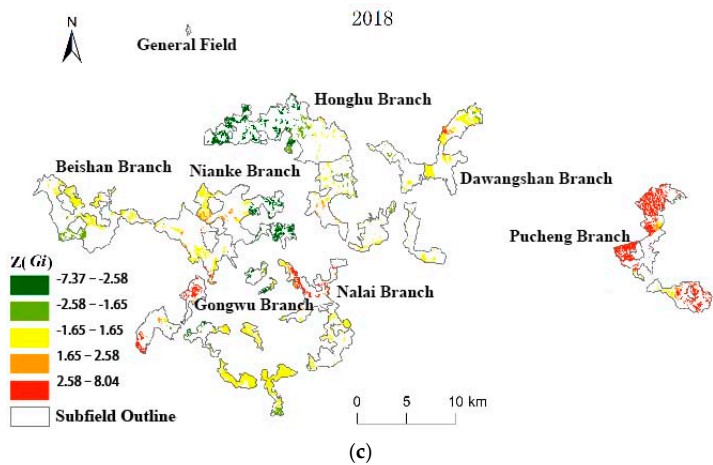

**Figure 10.** The distribution of cold–hot spots of soil conservation for PM plantations in Paiyangshan Forest Farm from 2009 to 2018 is presented in this figure. The aggregation index degree parameter is denoted by Z(Gi*). The color red represents hot spot regions, orange represents sub-hot spot regions, yellow indicates insignificant regions, dark green denotes cold spot regions, and light green indicates sub-cold spot regions. Panel (**a**) displays the distribution of cold–hot spots of soil conservation in 2009; panel (**b**) shows the distribution in 2013; and panel (**c**) depicts the distribution in 2018.

### 3.4. Multiple ESs Cold Hot Spot Analysis

The distribution and extent of cold–hot spots of individual ESs have been explored in detail earlier. The same service provides different amounts of value in different regions, and different regions can also provide multiple ESs. However, the regions provide different service capacities, both high and low, specifically reflected in this paper by the amount of service value per unit area [20]. Furthermore, the cold–hot spots of four ES value amounts are spatially superimposed to identify the cold–hot spots of multiple ESs in different regions.

The overall spatial distribution of the number of multiple hot spot ESs in Guangxi Paiyangshan Forest Farm over the years shows a pattern of high in the east and west and low in the middle (Figure 11). The gray small classes indicate that there are no hot spot services in the small mark. The western and northern part of Pucheng branch is the distribution area of four hot spot ES supply, while the southern part of Gongwu branch only provides one hot spot ES. Among these, the number of hot spot ESs provided by small classes of PM plantation forest in Gongwu branch is more complicated. In 2013, the number of hot spot ESs increased in Beishan branch and the southern part of Gongwu branch, while it decreased in the northwestern and eastern part of Gongwu branch. In 2018, the number of hot spot ESs decreased in the southern part of Gongwu branch and increased in Beishan branch, the central part of Gongwu branch, Honghu branch, Nalai branch, and Dawangshan branch. The largest number of small groups was able to provide three to four hot spot services from 2008 to 2019.

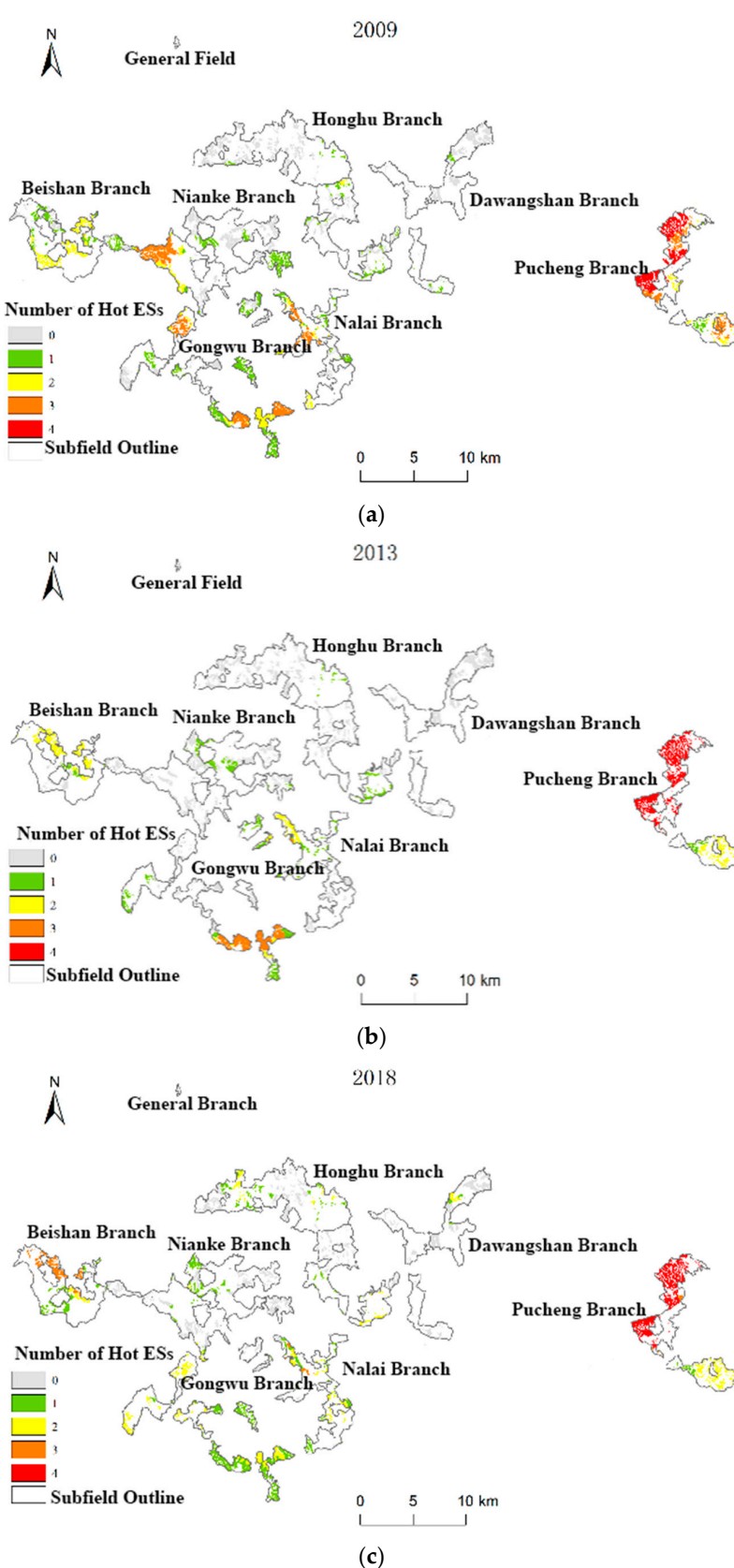

**Figure 11.** Number of hot spot ESs in PM plantations in Guangxi Paiyangshan Forest Farm from 2009 to 2018: (**a**) 2009; (**b**) 2013; (**c**) 2018.

## 4. Discussion

Most studies investigating the interconnections of various ESs have focused on administrative units such as provinces, basins, watersheds, and counties, but few have examined the spatial associations between ES pairs at the scale of forestry units. In this study, we quantified the synergistic and trade-off relationships among four ESs—wood supply, carbon sequestration and oxygen release, water conservation, and soil conservation—in Guangxi Paiyangshan Forest Farm, and explored the spatial autocorrelation distribution and hot spot distribution of ES pairs. Correlation analysis can provide a good overall perspective for qualitatively analyzing trade-off or synergistic relationships of ES pairs, but it cannot fully capture the spatial information of these relationships. Therefore, spatial autocorrelation analysis is a good complement to the analysis. Our results indicate that the relationships among ESs in PM plantations, like other forest ecosystems [34,35], exhibit a variety of synergistic and trade-off relationships, with obvious spatial heterogeneity and a predominance of synergistic relationships.

Different spatial scales can influence the interactions among ESs, and the synergies and trade-offs among them can also change [36]. By comparing the correlation coefficients among the four ES functions in PM plantations in Guangxi Paiyangshan Forest Farm, we identified the important relationships of ESs at the camping unit scale. We detected a significant positive correlation between the ES functions of the PM plantation, indicating a strong synergistic relationship between wood supply, carbon sequestration and oxygen release services, and a weak synergistic relationship between wood supply, water conservation, and soil conservation, which is consistent with the findings of Dai, E. et al. [6]. We also observed a weak synergy between water conservation services and the other three services (water conservation and soil conservation increased from weak to moderate synergy). Carbon sequestration and oxygen release had a weak synergistic relationship with water conservation, and carbon sequestration and oxygen release had a weak synergistic relationship with soil conservation services. Furthermore, we identified an important trade-off between carbon sequestration and oxygen release services and water conservation services. Based on the previously conducted ES assessment of PM plantation forests [20], it is known that the wood supply capacity, as well as carbon sequestration and oxygen release capacity, of PM plantation forests increase with the accumulation of stand volume and productivity with the increase in forest age. At the same time, soil erosion slows down under the effect of vegetation cover, and the soil conservation capacity of PM plantation forest increases. However, the water conservation capacity is mainly influenced by both rainfall and evaporation, with the increase in vegetation cover, evaporation increases and water conservation capacity decreases. Therefore, forest management should focus on the holistic theory of ESs and balance the relationship between provisioning services and regulating and supporting services according to local environmental problems to improve the overall benefits, rather than the economic benefits of a single service.

Overall, the ESs of PM plantation forests are dominated by synergistic relationships. There is a strong synergy between provisioning services and regulating services (carbon sequestration and oxygen release), a weak synergy between provisioning services and regulating services (water conservation), a weak synergy between provisioning services and supporting services, and a weak synergy between regulating services and supporting services. This indicates that the presence and growth of PM plantations contribute to the positive growth of these three services, providing a basis for multi-objective management operations in PM plantations. However, if the wood supply capacity is realized solely as timber output capacity, it will create a trade-off with the other three services. The output of timber volume will weaken the stand stock and ultimately affect the ESs of the whole stand [37,38].

There are four types of spatial relationships among ES pairs in PM plantations: "HH" synergistic relationship, "LL" synergistic relationship, "HL" trade-off relationship, and "LH" trade-off relationship. Understanding the overall synergy and trade-offs is important,

but it is also necessary to understand the spatial distribution of these relationships in order to develop strategies that can be tailored to local conditions.

The analysis of ES hot and cold spots in PM plantations revealed that the relationship between ES pairs affects the differences in service provisioning capacity among small groups of plantation forests in the region. The probability of multiple service hot spots occurring in small classes with synergistic changes in service pairs is higher, and their spatial distribution is increasingly similar. This finding is consistent with the results of Tian, Y.'s study [26]. On the other hand, the hot and cold spots provided by small plantation classes showing trade-off relationships overlap more, and the likelihood of hot spots appearing in the same area is lower, resulting in greater spatial variability. Furthermore, while some small plantation classes may provide multiple hot spot services, others may only provide one or no hot spot services.

### 5. Conclusions

This study utilized correlation analysis, bivariate spatial autocorrelation, and hot spot analysis to synthesize the synergistic and trade-off relationships over time and space of ESs in PM plantations in Guangxi Paiyangshan Forest Farm from 2009 to 2018, and revealed the following main findings:

- There were significant correlations among the four pairs of ESs in Guangxi Paiyangshan Forest Farm. A strong synergistic relationship was observed between wood supply and carbon sequestration and oxygen release services, whereas a weak synergistic relationship existed between wood supply and both water conservation and soil conservation. Additionally, a weak synergistic relationship was observed between water conservation and the other three services. Notably, the synergistic relationship between water conservation and soil conservation improved from weak to moderate, while the synergistic relationships between carbon sequestration and oxygen release and both water conservation and soil conservation were weak. Overall, supply services had a strong synergy with regulating services (carbon sequestration and oxygen release), a weak synergy with regulating services (water conservation), a weak synergy with supporting services, and regulating services had a weak synergy with supporting services.
- There were significant spatial differences in the bivariate spatial autocorrelations among the four ES pairs, and the overall spatial trade-off and synergistic relationships among the service pairs were generally consistent with the results of the correlation analysis. Wood supply showed synergistic relationships with carbon sequestration and oxygen release, water conservation, and soil conservation, with the strongest synergistic relationship observed between wood supply and carbon sequestration and oxygen release services. The synergistic relationship between wood supply and other services weakened over time.
- The spatial distribution of cold–hot spots for each ES in Guangxi Paiyangshan Forest Farm varies. There are both similarities and significant differences in the spatial distribution of cold–hot spots of different types of ESs. The distribution of cold–hot spots for provisioning services and regulating services (carbon sequestration and oxygen release) was similar, while the distribution of cold–hot spots for carbon sequestration and oxygen release, as well as water conservation services, which are also regulating services, differed significantly.
- Once the trade-offs and synergistic relationships between service pairs in time and space are understood, the decision-making process for PM plantation management should consider the spatial heterogeneity of different ecological processes and their links to different services. It is important to recognize the interrelationships among ESs, clarify their spatial and temporal characteristics, and identify the driving mechanisms behind them. Coordination and optimization of multiple services should be prioritized, and one service should not be enhanced at the expense of others.

- This study has some limitations. For example, while analyzing the trade-offs and synergies of ESs for the entire PM plantation in Guangxi Paiyangshan Forest Farm, the study lacked research on the scale effects of trade-offs and synergies of ESs [39]. The research on the value of ESs and the factors influencing the synergy and trade-off relationship between multiple services is not deep enough, and the study only focused on the ecosystem supply, regulation, and support services while neglecting the cultural services of PM plantations. In the future, we will continue to study the synergy and trade-off relationship between multiple ecosystem services in depth.

**Author Contributions:** R.M. and Y.M. conceptualized the framework, acquired the funding and supervised the overall project; L.L. and Y.W. collected the data; R.M. and Y.M. analyzed the data; R.M. and Y.M. wrote the manuscript; J.M., L.L. and Y.W. provided modification comments; J.M., L.L. and Y.W. reviewed the final manuscript. All authors have read and agreed to the published version of the manuscript.

**Funding:** This work was supported by the National Natural Science Foundation of China (32260387), the Open Research Fund of Guangxi Key Laboratory of Superior Timber Trees Resource Cultivation (2019-B-04-01), the Guangxi Key Research and Development Program (GuikeAB21220057), and the Guangxi Innovation-Driven Development Project (GuikeAA20161002-1 and GuikeAA17204087-7).

**Institutional Review Board Statement:** Not applicable.

**Informed Consent Statement:** Not applicable.

**Data Availability Statement:** The data are contained within the article.

**Acknowledgments:** We thank Zhangqi Yang of Guangxi Forestry Science Research Institute for his guidance on experimental methods, the state-owned Paiyang Mountain Forest Farm of Guangxi Zhuang Autonomous Region for help in field sampling, and graduate students from the School of Life Sciences of Guangxi Normal University for their assistance in indoor experiments.

**Conflicts of Interest:** The authors declare no conflict of interest.

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
