# Peer review of "The Trade-Offs and Synergies of Ecosystem Services in Pinus massoniana Lamb. Plantations in Guangxi, China"

_forests, doi:10.3390/f14030581_

Round 1

Reviewer 1 Report

The aim of this study was to quantified the synergistic and trade-off relationships among four ecosystem services, namely  wood supply, carbon sequestration and oxygen release, water conservation and soil conservation in Guangxi Paiyangshan Forest Farm, China, and to explore the spatial autocorrelation distribution and hot spot distribution of ES pairs. The present MS is a follow up of another recently published paper by the same Authors: “Ecosystem Service Evaluation and Multi-Objective Management of Pinus massoniana Lamb. Plantations in Guangxi, China” (Forests 2023, 14(2), 213; https://doi.org/10.3390/f14020213), which reports the calculation of the four ES whose synergies and trade-offs are investigated here.

The topic is fully within the scope of Forests, and it is also a very “hot” topic in research. However, the manuscript suffers from some major weakness, that should be addressed for publication in Forests:

-          The ms should be restructured in order to make more explicit reference to the previous work. At this regard, the publication of results already appearing in the previous paper (e.g. Table 1) should be avoided, since it represents plagiarism.

-          English language should be revised by a mother tongue speaking, expert, also the text is full of typos, that make the reading very difficult.

-          Figures numbers should be checked and corrected (i.e. Figure 2 pag 15 is Figure 6 and so on Figures 7, 8, 9…). Please also check the reference numbering in the text.

-          References should be checked, a lor of errors are appearing in the text

-          All figure legends should be improved since they miss some important details to make the figures self-explanatory. In particular: Figure 2: Please add in the legend the meaning of symbols (asterisks) and numbers; Figures 3-5: please explain the acronyms and colours used in the map; Figure 6-9 Please report here also the meaning of Z(Gi) and of the class values/colours reported in the map.

-          Discussion is quite poor and descriptive, mostly repeating already illustrated results. It should be improved considering recent available literature on forest ES synergies and trade-offs, and should be focussed on explaining the mechanisms of the observed synergies and trade-offs in ES.

-          Conclusion is not a summary of the work! Please state if the experimental hypothesis is verified, the unsolved problems and the future perspectives.

Author Response

I have revised the manuscript as you suggested, so please do not hesitate to contact me if you have any further questions.

Reviewer 2 Report

Dear authors, 

thank you so much for a very interesting and relevant study of synergistic and trade-off relationships among ecosystem services of Masson pine plantations in Guangxi.

Please, find below some remarks and questions:

Line 33-34. You should improve keywords. The title and keywords should not have the same words. 

L. 87 – Why do you use hm2?

L. 100 – You can delete h after 1650.3

L. 110 – Error! Reference source not found.

L. 111-112 – Could you add short definitions of age groups?

Table 1:

– Is it the result of your study? Why did you put table 1 in “Study Area”?

– How did you get it? How did you calculate the total Value (RMB) and the value per Hectare (RMB/hm2)? 

– What is RMB? Is it CNY?

– “Wood provided” – Do you mean wood of tree stems? Please, describe it.

– Why did you calculate together carbon sequestration and oxygen release?

– Please, add a reference to Table 1 to the text before Table 1.

L. 173 – Error! Reference source not found.

L. 183-199 – Maybe the data present in Table.

Figure 2 – a, b, c figures can be bigger.

L. 237-241 – You can add acronyms for "high-high", "low-low", "low-high" and "high-low".

L. 248-252 – Error! Reference source not found.

L. 266 – Please, add space.

L. 267 – Please, describe what Wood Supply means in this study.

Please, add a reference to all Tables in the text of the article.

L. 274-275 – Error! Reference source not found.

L. 306-307 – Error! Reference source not found.

L. 381 – Error! Reference source not found.

L. 392 – Error! Reference source not found.

L. 392-415 – Please, improve this text.

L. 426 – Error! Reference source not found.

L. 426-453 – Please, improve this text.

L. 465 – Error! Reference source not found.

L. 465-490 – Please, improve this text.

L. 501-524 – Please, improve this text.

L. 540 – Error! Reference source not found.

L. 565 – Authors should improve the discussion. Please, add comparing your results with another study.

The weak side of the study is the use of analysis of primary data of unclear origin. How readers can understand a study result if they can not check the origin of incoming data?

Author Response

(The authors gave the same response as above.)

Reviewer 3 Report

The study is relevant and the subject is worthy of investigation.  However, some parts of the manuscript are confusing and there are numerous errors or omissions, which should be carefully reviewed by the authors.  Therefore, I recommend a moderate review of the document, if the authors incorporate the requested improvements, the article could be considered for publication in the journal.  The questions addressed by the authors are detailed below:

1. GENERAL  CONCERN

The document needs careful review, both from the point of view of language editing, and from the perspective of narrative structuring.  It is recommended that it be proofread by a native English proofreader or language editing service.  In addition, there are repetitive oversights such as the correct insertion of the bibliographical references (numerous error messages appear in the PDF document) or the incorrect numbering of the figures.  Therefore, authors are recommended to carefully review the document.

2. METHODOLOGY

The methodology section is excessively scarce and quite confusing.  A number of indicators of geostatistical analysis are briefly listed, but there is no comprehensive methodological framework.  Therefore, it is necessary that this detail which is the specific methodological framework, explaining the different phases that are going to be applied so that it is what is explained in the methodology section presents a clear correspondence with what is presented in the section of results.  In this sense, it is recommended to include a schematic summary of the proposed methodological process so that the reader is clear about the narrative thread of the methodology section.

3. LITERATURE REVIEW AND DISCUSSION 

The document in general presents an excessively local approach.  It would be necessary to raise its level of scientific academic scholarship by bringing a more international approach citing relevant non-Chinese case studies and research.  This would make the manuscript more interesting to a broader readership of the journal.  In addition, the scientific discussion section suffers from the lack of self-critical content that raises the limitations of the research carried out and what issues could be improved in future lines of research.

Author Response

(The authors gave the same response as above.)

Round 2

Reviewer 1 Report

The paper has been improved, however in my opinion Authors should better highlight in the text, specifically in the aims paragraph, that they are performing their analyses using the data published in the previous paper by Mo et al., 2023.

Similarly, also the description of the study area, being the same of the previous study, should explicity cite the previoous paper.

There are still some error warnings in the text that should be corrected (e.g. line 432).

Author Response

I have changed it as requested.

Reviewer 2 Report

Dear authors, thank you for your efforts and for improving the manuscript. 

Please, check the parentheses in Figure 3.

Author Response

I have checked the comments in Figure 3 and changed the errors.